# Ocean dynamics shape marine heatwaves and their predictability

Xianglin Ren[1], Wei Liu [1] ✉ & Liping Zhang [2,3]

Marine heatwaves have become more frequent and intense under anthropogenic warming, posing increasing threats to marine ecosystems and coastal societies, necessitating a better understanding of their mechanism and predictability. Here we show how ocean dynamics modulate marine heatwaves globally by comparing dynamic and slab ocean climate model simulations. We discover that ocean dynamics significantly promote marine heatwave intensity and duration in mid-to-high latitude oceans, as well as the eastern tropical Pacific where marine heatwaves are inherently linked to extreme El Niño events. Our mixed-layer heat budget analysis unravels that heat accumulation during marine heatwave episodes is strongly influenced by vertical mixing and horizontal transport processes, so that warm sea surface temperature extremes in dynamic ocean differ in magnitude and evolution rhythm from those in slab ocean. We further find robust multi-year potential predictability of marine heatwave in the North Atlantic with a dynamic ocean, owing primarily to the predictability of the Atlantic Meridional Overturning Circulation. Our findings emphasize the irreplaceable role of oceanic dynamics in marine heatwave evolution and predictability, with important implications for future climate extreme prediction and adaptation strategies.

Marine heatwaves (MHWs), defined as prolonged periods of anomalously warm sea surface temperatures (SSTs), have occurred frequently in recent decades, posing serious threats to marine ecosystems and coastal communities[1–4]. These extreme warm events have disrupted marine food webs and impacted fisheries[5,6], triggered widespread coral bleaching[7], and driven large-scale ecosystem restructuring[8,9]. With continued global warming, MHWs are projected to become more frequent, intense, and widespread[1–3,10]. Understanding their drivers and improving their predictability has become a central challenge in climate science.

Previous studies suggested that anthropogenic climate change, particularly the accumulation of greenhouse gases, will elevate the global temperature baseline, thereby enlarging the likelihood of MHWs[1,11,12]. Climate variability, like the Pacific Decadal Oscillation, have dramatically influenced the MHWs in the North Pacific[2,13–15]. Atmospheric circulations associated with the North Atlantic Oscillation, manifesting blocking high-pressure systems, can affect MHW evolution by altering surface wind patterns and air-sea heat fluxes[2,10]. Beyond these drivers, ocean dynamics play an indispensable role in modulating MHWs through mechanisms such as anomalous warm current and weakened coastal upwelling. For example, the 2011 Ningaloo Niño and 2015/16 Tasman Sea MHW events were suggested to associate with a stronger Leeuwin Current, while reduced cold-water upwelling made surface warming persistent along the California coast and in the Mediterranean Sea[16–18]. Of particular interest is the Atlantic Meridional Overturning Circulation (AMOC), which regulates the upper ocean heat content and mixed layer depth in the North Atlantic, providing a critical physical foundation for MHW formation in this region[19]. Given its central role in meridional heat transport and its decadal variability[20,21], the AMOC may even serve as an important yet underexplored source of MHW predictability, from multiyear to decadal timescale.

It is apparent that most of the existing studies focused on individual events or isolated mechanisms. Despite different types of El

[1]Department of Earth Sciences and Planetary Sciences, University of California Riverside, Riverside, CA, USA. [2]NOAA/Geophysical Fluid Dynamics Laboratory, Princeton, NJ, USA. [3]University Corporation for Atmospheric Research, Boulder, CO, USA. ✉e-mail: wei.liu@ucr.edu

Niño-Southern Oscillation (ENSO) events were suggested to affect global MHWs via teleconnection in which dynamical ocean processes could be involved[22], the contribution of ocean dynamics to MHW evolution on a global scale still needs to be explicitly and systematically evaluated. More importantly, the associated MHW predictability has yet to be studied over a global scale.

In this work, we will compare slab ocean and fully coupled simulations with the same model to identify the effects of ocean dynamical processes on MHW frequency, intensity, and duration across different regions over the globe. We will further assess whether AMOC predictability renders the decadal predictability of MHWs in the North Atlantic.

## Results

### Distinct MHWs in distinct regions

We leverage the Community Earth System Model version 1 (CESM1) preindustrial slab ocean and fully coupled (dynamic ocean thereafter) simulations (Methods) and compare global MHW duration, annual frequency, and intensity between the two simulations (Fig. 1). We adopt daily and monthly MHW definitions (Methods), both of which result in consistent MHW patterns (Fig. 1 and Supplementary Fig. 1). One of the most striking features emerges in the eastern tropical Pacific where MHW duration and intensity significantly enhance in dynamic ocean, by 45% and 47%, respectively, when compared to slab model. Such MHW feature relates to ENSO activity in this region[23,24]. Here, we display the Niño3.4 SST standard deviation and 2–7-year ENSO power spectra in both simulations, which are 31% and 59% larger in dynamic than slab ocean (Fig. 2a–d), owing primarily to the dynamic coupling between tropical ocean and atmosphere and upper ocean dynamics[25–27]. In dynamic ocean, ocean dynamics participate in controlling the evolution of ENSO events by regulating the positive Bjerknes feedback and the delayed negative feedback, meanwhile, ocean heat transport and vertical mixing processes are more realistically represented in dynamic ocean, supporting the persistence of SST

anomalies[28,29]. In slab ocean, however, SST anomalies strongly rely on local surface heat flux, atmospheric feedback, and ocean heat capacity[25]. It is worth noting that the aforementioned MHW difference between dynamic and slab oceans in CESM1 is consistent across a broad suite of climate models (Supplementary Fig. 2).

We further elucidate the role of ocean dynamics in MHWs in the tropical Pacific through the lens of extreme ENSO. In comparison to normal ENSO, extreme ENSO has larger magnitude in all feedbacks involved in the Bjerknes stability (Methods), while its thermocline feedback has the largest positive contribution to the stability, thus serving as the key factor in raising its Bjerknes stability over than that of normal ENSO (Supplementary Fig. 3). Benefited from the monthly MHW definition, MHWs in the region are intrinsically linked to extreme ENSO events that are characterized by unusual high SSTs with Niño 3 precipitation exceeding 5 mm/day[30]. We analyze the relationship between winter SST and precipitation in the Niño 3 region, comparing dynamic and slab ocean conditions (Fig. 2e, f). In contrast to slab ocean, we see more frequent extreme ENSO episodes in dynamic ocean, accompanied by stronger tropical precipitation anomalies. This indicates that ocean dynamics can exacerbate extreme ENSO occurrence by promoting ocean-atmosphere feedback and vertical oceanic heat redistribution, thus leading to stronger and more persistent MHWs in the tropical Pacific[4,31].

Beyond the tropics, we find that ocean dynamics also play a significant role in modulating MHWs in mid- to high latitudes. A dynamic ocean allows for more frequent and shorter-duration MHWs than a slab ocean over the North Pacific, North Atlantic and Southern Ocean, especially to the south of Greenland and in the Labrador Sea (Fig. 1c, f). The existence of ocean dynamical process also reduces the intensify of MHWs in the Northeast Pacific and Northwest Atlantic (Fig. 1i). Unlike ENSO dynamics, accounting for the MHW differences between dynamic and slab oceans in these regions is more complex than in the tropical Pacific, hence we resort to a regional mixed-layer heat budget analysis.

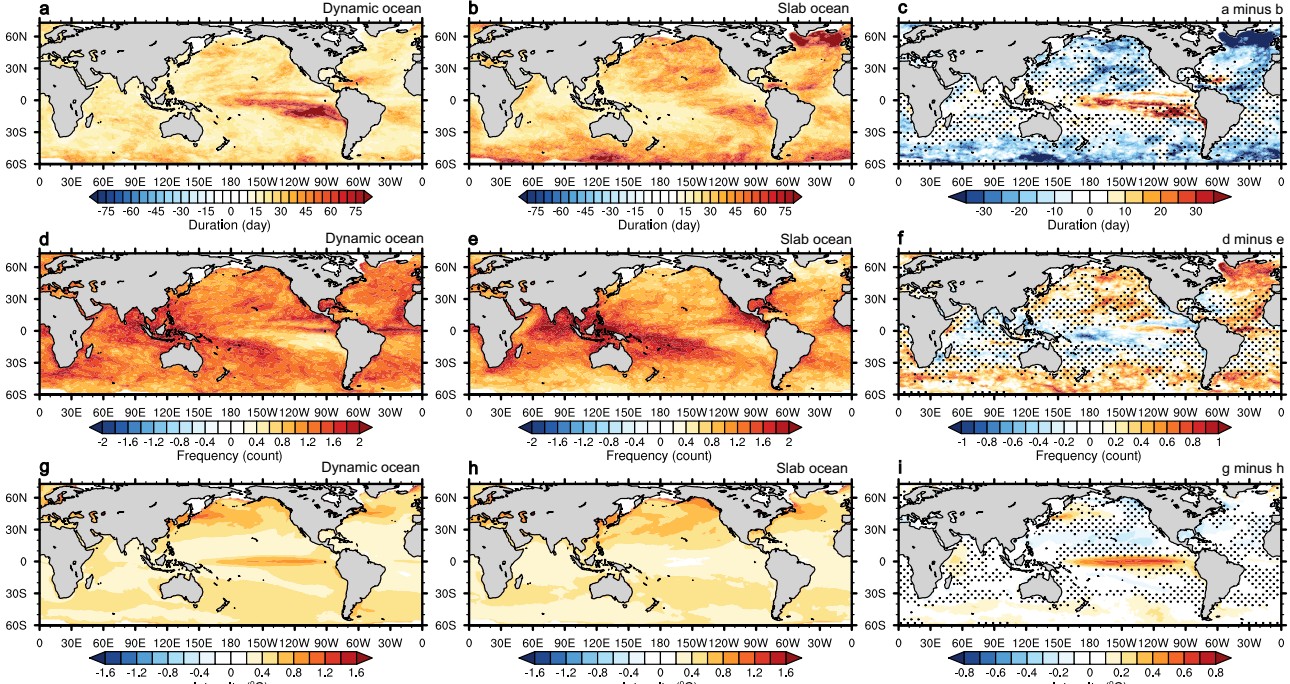

**Fig. 1 | Marine heatwaves in dynamic and slab oceans.** Marine heatwave durations (color shading in days) in (**a**) dynamic and (**b**) slab oceans based on a daily definition, as well as (**c**) the difference between the two (dynamic minus slab). **d–f** Same as (**a–c**) but for marine heatwave annual frequencies. **g–i** Same as (**a–c**) but for marine heatwave intensity. The stipples in panels (**c, f, i**) refer to the regions where differences are statistically insignificant based on Student's t-test at the 95% confidence level. The base map is from NCAR Command Language map outline databases.

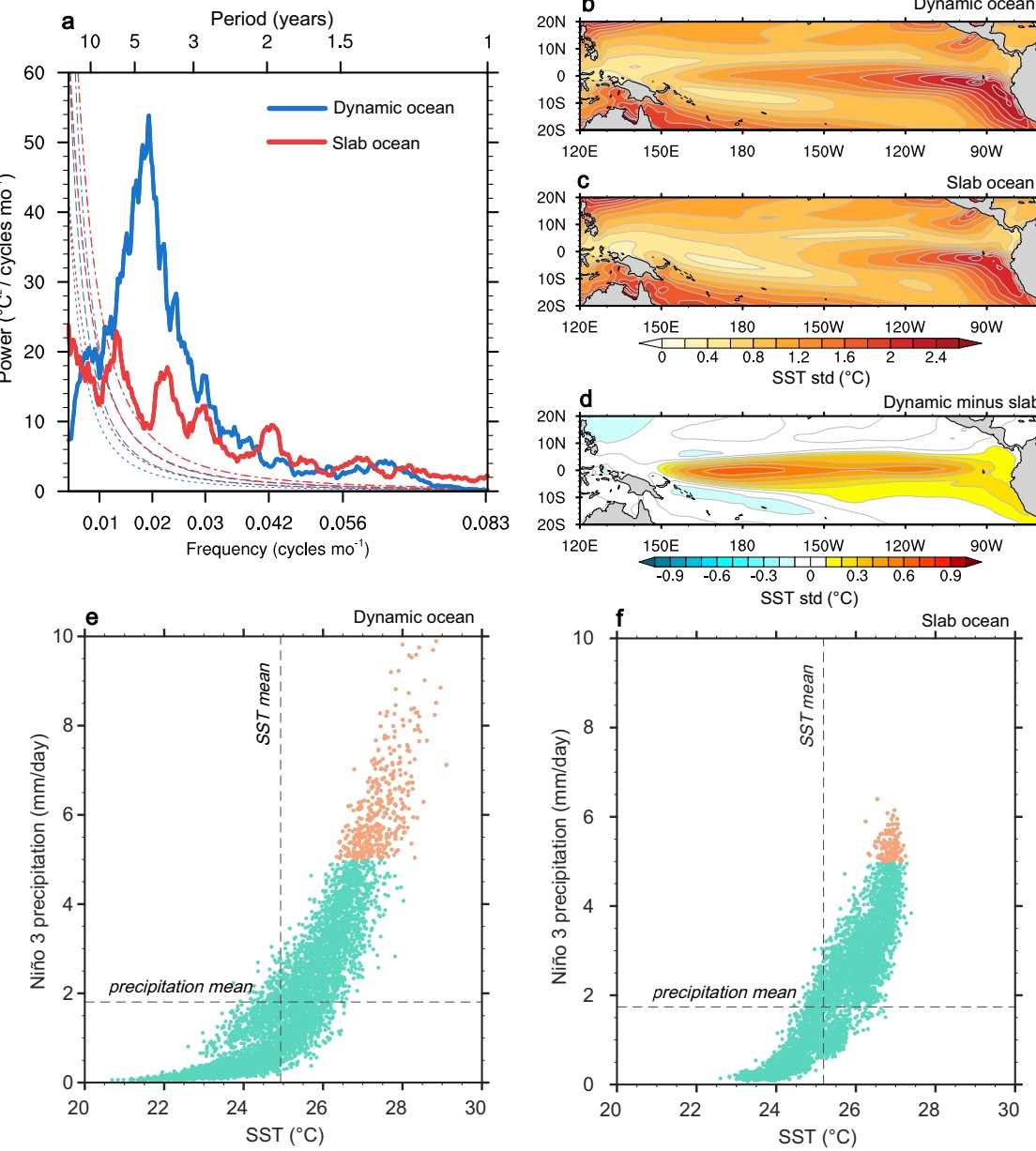

**Fig. 2 | El Niño-Southern Oscillation in dynamic and slab oceans. a** Power spectra of the Niño 3.4 indices of dynamic (blue) and slab (red) ocean simulations as well as their 95% confidence limits (dashed/dotted curves). Sea surface temperature standard deviations (color shading in °C) for (**b**) dynamic and (**c**) slab oceans, and (**d**) the difference between the two (dynamic minus slab). Scatterplots of monthly sea surface temperature versus monthly precipitation in the Niño 3 region for (**e**) dynamic and (**f**) slab oceans. Orange dots represent the cases when precipitation is larger than 5 mm/day. The black vertical and horizontal dashed lines denote the climatological mean values of sea surface temperature and precipitation, respectively. The base map is from NCAR Command Language map outline databases.

## Ocean dynamics modulating MHWs

We perform a mixed-layer heat budget decomposition to look at how the key components in the budget evolve throughout the course of MHW's lifecycle (Methods). We example four MHW hotspots over the past few decades[2,23], i.e., the Niño 3.4 region where MHWs intensify from a slab to dynamic ocean, and the Mediterranean Sea, Gulf of Alaska, and Gulf Stream regions where MHWs decline, to delve into the distinct impacts of ocean dynamics on MHWs (Fig. 3a). For each region, we make a composite of heat budget when MHWs peak (designated as month 0), as well as 6 months before and after the peak, to depict the evolution of the heat budget components during the lifecycle of MHWs (Fig. 3b–e).

Over the Niño 3.4 region, dynamic ocean exhibits a rapid heat accumulation 1–2 months before the MHW peak, followed by a decline afterward (Fig. 3b). By contrast, the heat budget components in slab

ocean have amplitudes only about one-fourth of those in dynamic ocean. This is because SST variations in slab ocean are mainly driven by variations in net surface heat flux (Supplementary Fig. 4a), lacking the modulation from internal ocean dynamical processes. In dynamic ocean, both surface heat flux (Supplementary Fig. 4a) and horizontal diffusion tendency remain positive over the whole period, though they could be offset by negative advective and vertical diffusion tendency, which collectively control the life cycle of MHWs (Fig. 3b). Cold upwelling in the eastern equatorial Pacific induces a negative vertical diffusion tendency, persistently exerting a cooling effect on SST anomalies[31]. Meanwhile, under the influence of prevailing easterly winds, warm surface water is transported westward and away from the Niño 3.4 region, producing a negative advective tendency. In addition to this horizontal advection, the strengthened easterlies also enhance equatorial divergence and vertical upwelling, both of which further

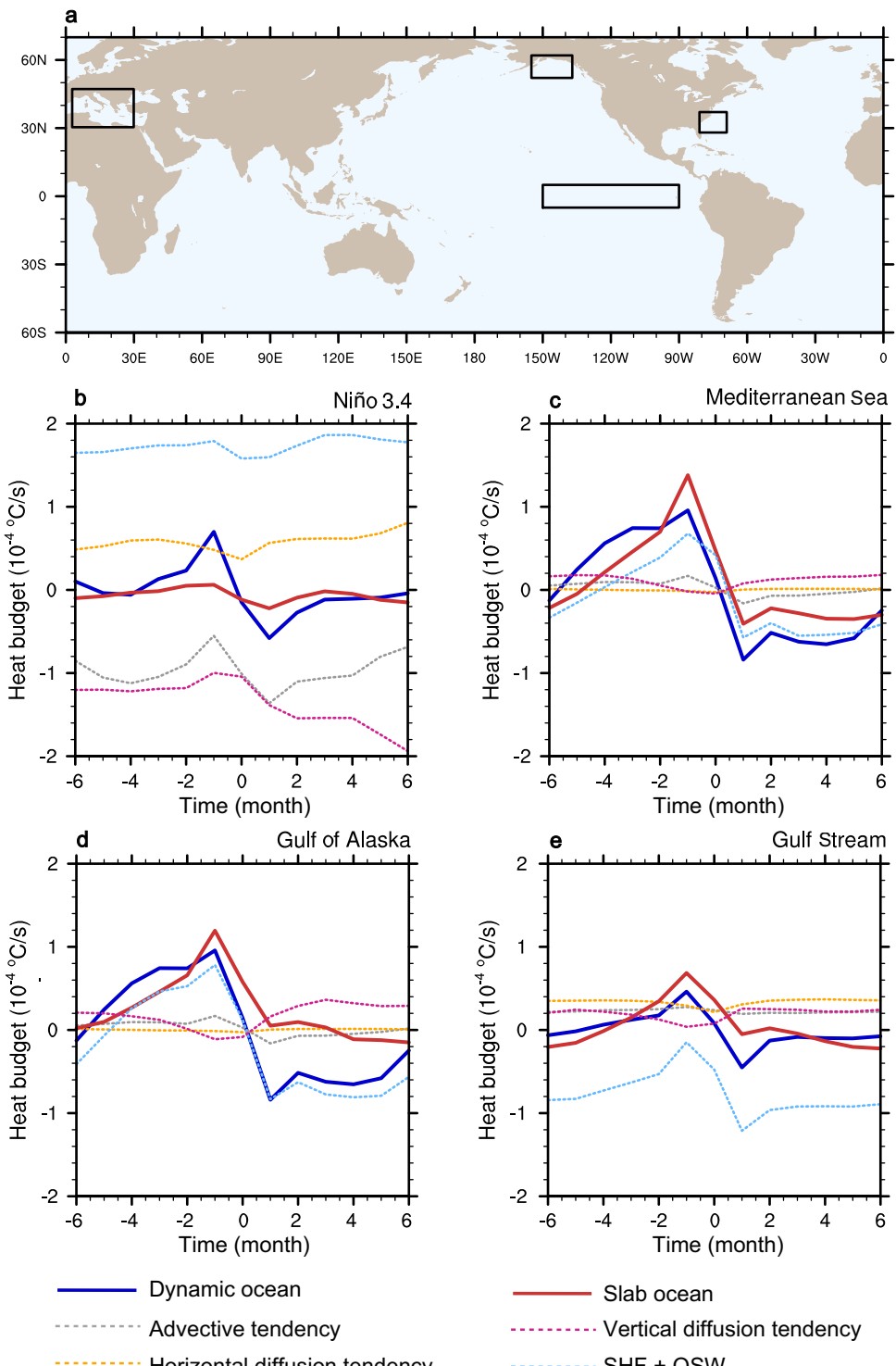

**Fig. 3 | Mixed-layer ocean heat budgets during marine heatwave events.** Evolution of regionally averaged mixed-layer ocean heat budget components composited relative to marine heatwave peak (month 0) in four boxed regions in panel (**a**), i.e., the (**b**) Niño 3.4, (**c**) Mediterranean Sea, (**d**) Gulf of Alaska, and (**e**) Gulf Stream regions. Solid lines represent the total temperature tendencies for dynamic (blue) and slab (red) oceans. Dotted lines represent the decomposed components of heat budget in dynamic ocean: advective tendency (gray), horizontal diffusion tendency (orange), vertical diffusion tendency (magenta), and net surface heat flux (SHF + QSW, light blue, downward positive). The base map is from NCAR Command Language map outline databases.

contribute to local SST cooling[32]. Although the advective term is negative on average, it exhibits a distinct temporal evolution throughout the MHW lifecycle (Fig. 3b). During the early stage of the event, westerly wind bursts in the equatorial warm pool may temporarily increase eastward heat transport, weakening the advective

cooling. In the later stage, as easterlies recover, westward advection strengthens, and the advection component becomes more negative.

In contrast to the eastern tropical Pacific, MHWs are weaker in dynamic than slab ocean over the Mediterranean and Gulf of Alaska regions. The SST extreme in slab ocean is driven by net surface heat

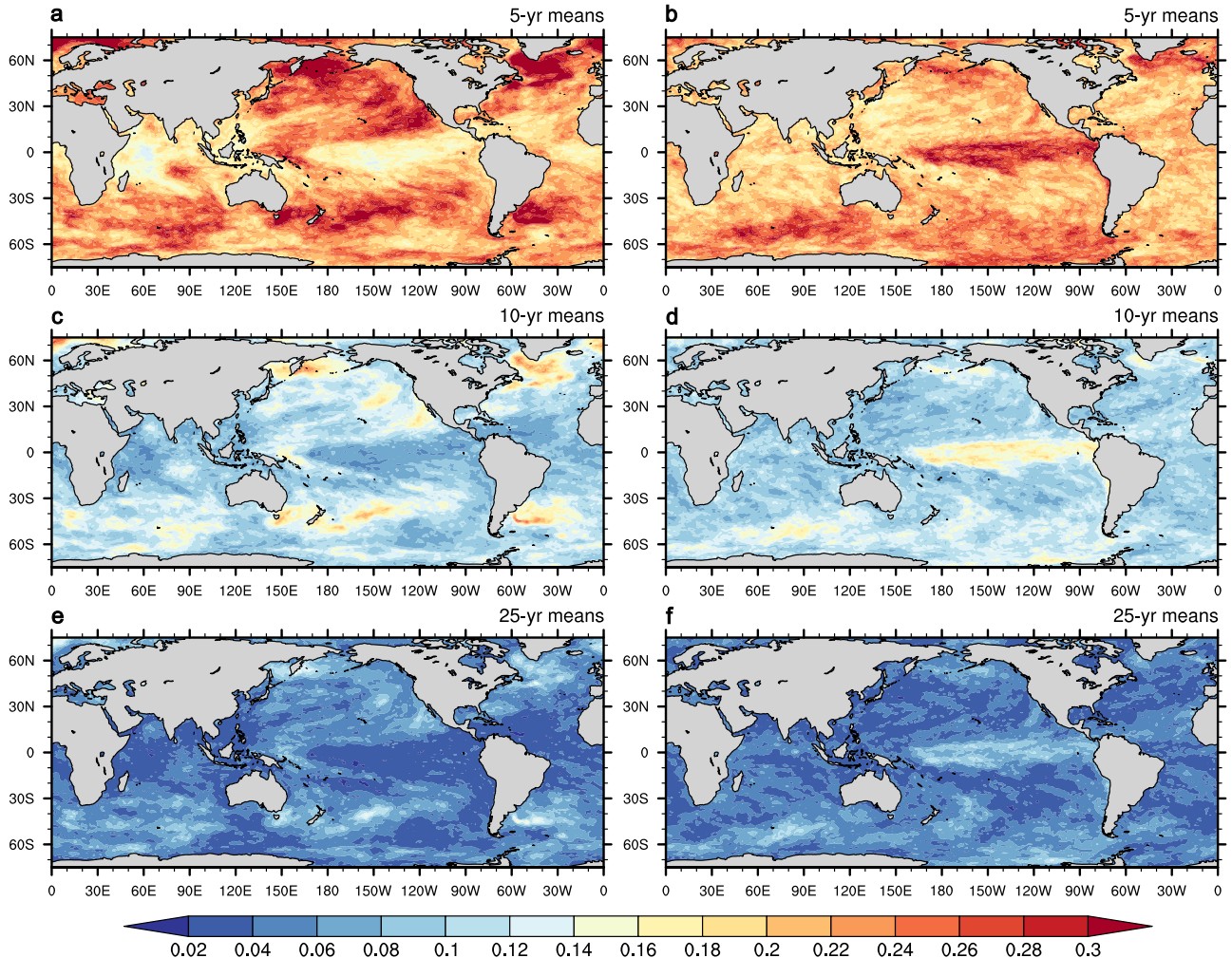

**Fig. 4 | Potential predictability variance fraction for marine heatwave frequency.** Potential predictability variance fraction (ppvf) for (**a**) 5-year, (**c**) 10-year, and (**e**) 25-year average marine heatwave annual frequency in dynamic ocean. **b**, **d**, **f** Same as (**a**, **c**, **e**) but for slab ocean. The base map is from NCAR Command Language map outline databases.

flux variation, which imposes a strong, short-term warming prior to the event peak (Supplementary Fig. 4b–c). However, in dynamic ocean, the warming effect from net surface heat flux is relatively weaker, and vertical mixing in the ocean acts as a buffer during the warming phase, lowering the magnitude of SST anomalies (Fig. 3c, d).

The Gulf Stream region deserves attention because its heat budget amplitude in dynamic ocean is much smaller than that in slab ocean, due mainly to a persistent negative net surface heat flux (Fig. 3e and Supplementary Fig. 4d). As one of the most notable regions for heat release from ocean to atmosphere via turbulent latent and sensible heat fluxes[33], even oceanic advection and diffusion processes bring about positive heat anomalies prior to MHW peaks, they are typically overwhelmed by strong surface heat loss (Fig. 3e). Therefore, it is difficult for heat to accumulate effectively in this region before and after the MHW peak, which limits the intensity of warm SST extreme.

**AMOC impact on North Atlantic MHW predictability**
Our heat budget analysis highlights the importance of ocean dynamics in regulating MHW evolution, and meanwhile, encourages the investigation into whether the differences between dynamic and slab oceans manifest as long-term, predictable signals within the climate system. To this end, we employ the potential predictability variance fraction (ppvf) as a diagnostic metric to evaluate the long-term

predictability of global MHW characteristics from the perspectives of spatial distribution and temporal scale.

Figure 4 depicts the spatial distribution of the ppvf of global MHW frequency under 5-year, 10-year, and 25-year running means. Relative to slab ocean (Fig. 4b, d, f), dynamic ocean (Fig. 4a, c, e) shows generally higher ppvf values in the North Atlantic, Southern Ocean, and western North Pacific, indicative of relatively stable low-frequency variability in these regions and hence potential predictability on multi-year to decadal timescale. This finding is consistent with previous studies on SST potential predictability[34,35].

By contrast, slab ocean displays considerably higher ppvf levels than dynamic ocean over the equatorial Pacific across all the timescales (Fig. 4b, d, f). Despite lacking active ocean dynamics such as horizontal heat transport, overturning circulation, and equatorial upwelling, the slab ocean model exhibits pronounced decadal-to-multidecadal SST persistence in the tropical Pacific. The enhanced low-frequency stability arises in that the shallow mixed layer acts as an integrator of persistent atmospheric forcing, particularly cloud radiative feedback[36] and wind-evaporation-SST (WES) feedback[37], without the counteracting oceanic negative feedback (e.g., heat redistribution via upwelling, Kelvin/Rossby waves) that is present in the fully coupled model. As a result, tropical SST anomalies in slab configurations can remain coherent across multi-decadal timescales[25,26,38,39].

In dynamic ocean, the North Atlantic has conspicuously high predictability in terms of MHW frequency, duration, and intensity across all the timescales, which is potentially linked to AMOC predictability. This is because the AMOC serves as a key factor controlling the variability of ocean heat content and SST in this region on a decadal scale[20]. We therefore use the Average Predictability Time (APT) approach (Methods) to identify the leading and the most predictable MHW frequency, duration, and intensity components over the North Atlantic as well as their linkage to the AMOC.

The most predictable component (APT1) for MHW frequency in dynamic ocean manifests opposing anomalies between the Labrador Sea/South Greenland and the Gulf Stream region (Fig. 5a). Such dipole-like pattern resembles the well-known AMOC fingerprint[20,40,41], indicating that the APT1 mode is potentially linked to low-frequency variability of the AMOC[42–44]. Seen from the squared multiple correlation coefficient ($R^2$, Methods), APT1 retains predictive skill up to around 5 years in advance (Fig. 5c), which could be understood from lead/lag correlations between AMOC and APT timeseries (Fig. 5d). In particular, the AMOC stream-function delineates a significant negative correlation with the APT1 index across the basin when the AMOC leads by 6 years, suggesting that it will take 6 years for a slowed AMOC to modify the SST over the North Atlantic. This is due primarily to the southward propagation of AMOC and associated meridional heat transport anomalies from the North Atlantic deep water formation region, at a speed of tracer advection along the interior pathways of the deep water[45], creating negative SST anomalies to the south of Greenland via reduced northward heat transport[39,46] and positive SST anomalies via shifting the Gulf Stream northward[43,47]. The distinct mean-state SST shifts in the two locations, in turn, alter their shapes of SST probability distributions differently, lowering the likelihood of MHW occurrence to the south of Greenland[19] but raising it in the Gulf Stream region. Our result explains that APT1 not only has statistical stability but also carries the heat memory of AMOC's long-term variability.

In slab ocean, the spatial structure of APT1 for MHW frequency is displaced eastward with smaller amplitude than dynamic ocean (Fig. 5b). The eastward shift likely results from an absence of the AMOC-driven western boundary intensification of heat transport and deep-ocean heat exchange under the slab ocean configuration. SST anomalies are more readily to form and persist in regions such as the west coast of Europe. Compared to dynamic ocean, the predictability time of APT1 is remarkably reduced (Fig. 5c), with a rapid decay of $R^2$ signal within 2 years.

In addition to MHW annual frequency, we also examine ppvf distributions (Supplementary Fig. 5 and Fig. 6), APT1, and the link between APT modes and the AMOC (Supplementary Fig. 7 and Fig. 8) for MHW duration and intensity. We find that the results are quite similar to those from MHW frequency, suggesting that AMOC predictability supports the multi-year potential predictability, not only for MHW frequency, but also for MHW duration and intensity, over the North Atlantic.

## Discussion

In this study, we compare the dynamic and slab ocean configurations with a broadly used climate model to assess the role of ocean dynamics in regulating the characteristics and predictability of global MHWs. We find that ocean dynamics significantly increase the intensity and duration of MHWs in multiple key regions, particularly in the eastern tropical Pacific where ENSO is highly active. SST variance and ENSO power spectrum with a 2–7-year cycle are much higher in dynamic than slab ocean, reflecting the importance of air-sea coupling feedback and ocean heat transport in maintaining strong SST anomalies. While ocean dynamics reduce MHW predictability in the ENSO region on interannual-to-decadal timescales. Our heat budget analysis shows that the accumulation of heat in MHW episodes in dynamic ocean is

markedly influenced by vertical mixing and horizontal transport processes, leading to SST anomalies that differ in amplitude and evolution rhythm from those in slab ocean. Furthermore, we find strong multi-year potential predictability in the MHWs over the North Atlantic with a dynamic ocean, owing primarily to AMOC predictability. For MHW frequency, duration, and intensity, we identify the most predictable mode. It reflects the heat transport regulation capacity of the AMOC at different stages, representing the lagged response of its positive and negative phases to MHW frequency, duration and intensity.

Our findings reveal that ocean dynamics not only shape the spatial structure and evolution of MHWs but also significantly enhance the multi-decadal predictability of MHWs in mid-to-high latitudes by modulating low-frequency SST variability. In particular, changes in the AMOC provide valuable predictive information for the North Atlantic. This AMOC induced predictability even persists under anthropogenic forcings (Methods, Supplementary Fig. 9). On the other hand, it merits attention that both ppvf and APT from the preindustrial simulations detect a perfect model skill, implying perfect initiation conditions and no model biases. As a result, the outcome reflects the upper limit of prediction skill. In reality, prediction skill is usually lower than perfect model skill due to the existence of model bias and imperfect initial conditions.

## Methods
### Observations
We employ the daily Optimum Interpolation Sea Surface Temperature version 2.1 (OISST v2.1)[48], which has a spatial resolution of 0.25°×0.25° and is available since 1 September 1981. Based on OISST v2.1, we illustrate the observed marine heatwaves (MHWs) over global oceans from 1982 to 2024. We also harness the monthly Extended Reconstructed Sea Surface Temperature version 6 (ERSSTv6)[49] that has a spatial resolution of 2°×2° and is available since 1850, as well as the monthly CPC Merged Analysis of Precipitation (CMAP)[50] that has a spatial resolution of 2.5°×2.5° and is available since 1979.

### Model simulation
We leverage the nominal 1-degree latitude/longitude version of the Community Earth System Model version 1 (CESM1) with CAM5 atmosphere and daily sea surface temperature (SST) output[51]. Specifically, we use two preindustrial control simulations: one with a fully coupled configuration (namely "dynamic ocean") and the other with a slab ocean configuration (namely "slab ocean"). The fully coupled configuration includes a dynamic ocean component (POP2) that explicitly resolves oceanic circulation and heat transport processes, while the slab ocean replaces ocean dynamics with a thermodynamic mixed-layer model constrained by prescribed Q-flux. Note that the slab ocean model adopts an annual mean mixed-layer depth throughout the year, so does not account for seasonal mixed layer variability, whereas MHWs could be associated with mixed layer depth variations[52]. We exclude MHWs when and where sea ice presents for either dynamic or slab ocean simulation, considering that surface temperatures in slab ocean represent the temperatures over the ice surface rather than the ocean surface. We exploit the last 500 years in either simulation, which are long enough to ensure robust statistical analysis of low-frequency variability, for example in dynamic ocean, the variability associated with the Atlantic Meridional Overturning Circulation (AMOC). Comparison between the dynamic and slab ocean simulations provides insight into how interactive ocean dynamics, along with coupled feedback processes, influence the evolution and predictability of global MHWs.

### Detection of marine heatwaves
MHWs are identified as periods of anomalously high SSTs exceeding a seasonally varying threshold. Following the definition established by ref. 53, we define an MHW as any period where SST exceeds the 90th

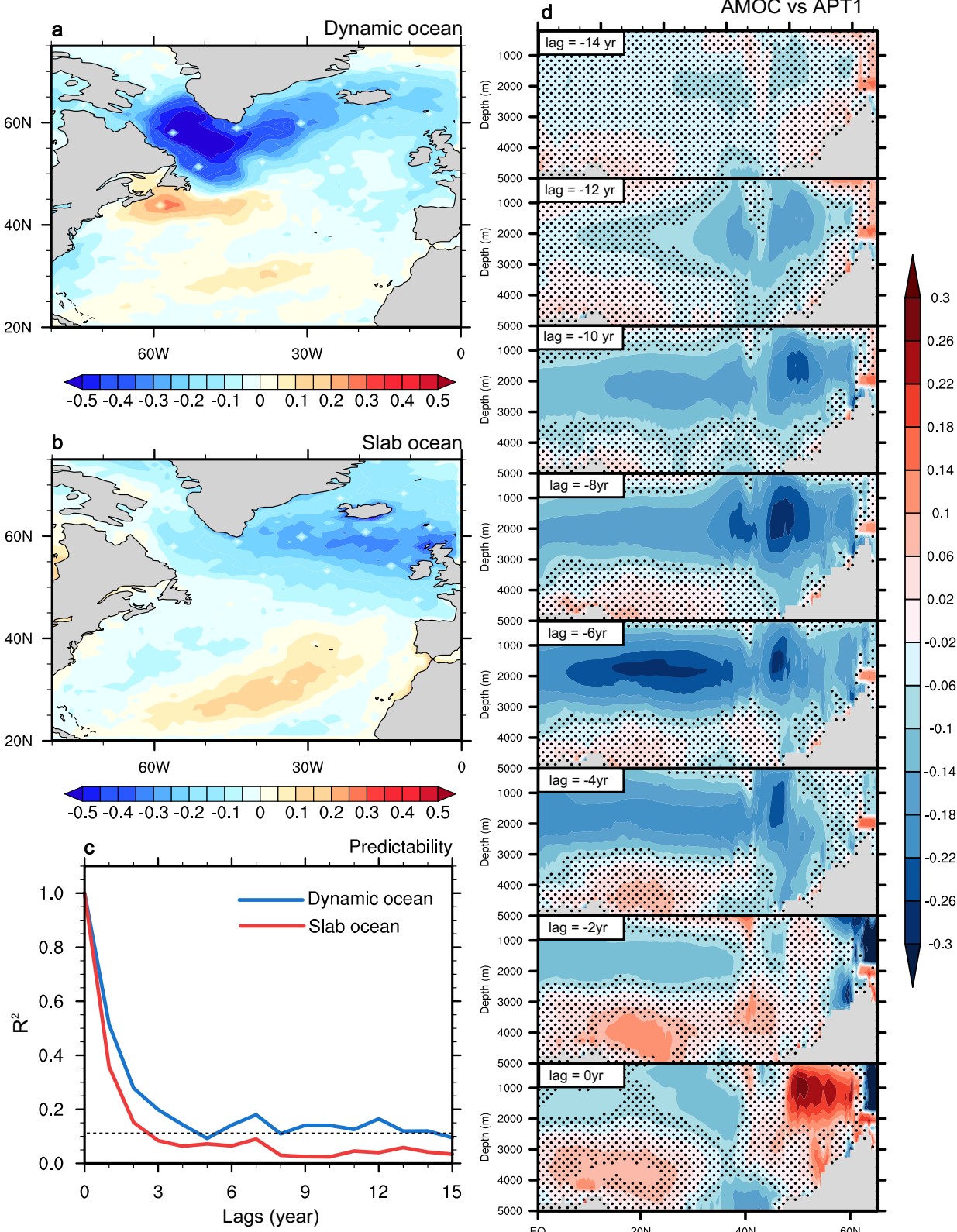

**Fig. 5 | North Atlantic marine heatwave frequency predictability in dynamic and slab oceans.** The leading predictable component (APT1) of North Atlantic marine heatwave annual frequency in (**a**) dynamic and (**b**) slab oceans, as well as (**c**) the squared multiple correlation coefficients $R^2$ (dynamic ocean, blue; slab ocean, red). The dashed black line in (**c**) denotes the 95% significance level. **d** Lead/lag correlation between Atlantic meridional overturning stream-function and APT1 index for North Atlantic marine heatwave annual frequency. Lags range from −14 to 0 years, in which negative lags indicate that the Atlantic Meridional Overturning Circulation (AMOC) leads the APT index. The stipples refer to the regions where correlations are statistically insignificant at the 95% confidence level. The base map is from NCAR Command Language map outline databases.

percentile of local daily climatology for at least five consecutive days. Two events with an interruption of less than 3 days are considered as one MHW event. For both simulations, the daily SST climatology and the corresponding 90th percentile threshold for each calendar day are calculated separately for the coupled and slab ocean runs, using the daily SST for all years within 11 days centered on that day, and the resulting time series are subsequently smoothed using a 31-day moving average. We analyze characteristics of MHWs such as frequency, duration, and intensity in CESM1 preindustrial control simulations over a 500-year period under dynamic and slab ocean configurations. Frequency is defined as the number of events per year; duration is defined as the time between the start and end date of the event; and intensity is defined as the maximum intensity of each MHW, denoting the maximum SST anomaly relative to a seasonally varying climatology over the duration of the event.

Besides the CESM1 dynamic-ocean preindustrial control simulation, we look into CESM1 large ensemble simulations of this configuration[54], which are under historical (by 2005) and representative concentration pathway 8.5 (after 2005) external forcings, consisting of 40 members between 1920 and 2024 (historical large ensemble for simplicity). We compare dynamic-ocean preindustrial MHWs with OISST v2.1 (1982–2024), and also with historical large ensemble MHWs during the same period (1982–2024) based on the daily MHW definition (Supplementary Fig. 10). We find consistent MHW characteristics amongst the three suites over a global scale. Nonetheless, as a recurrent issue in non-eddy-resolved models[55], CESM1 historical and preindustrial simulations have longer MHW durations, lower annual mean MHW frequencies and smaller MHW intensities than observations. Such model-observation discrepancy exists because in observation the daily SST climatology is based on this sole realization, whilst in the model, either a 40-member ensemble mean (for historical large ensemble) or a 500-year average (for preindustrial simulation) daily SST climatology can be harnessed. Comparison between CESM1 historical and preindustrial simulations further evinces that anthropogenic aerosol and greenhouse gas forcings seldom alter the global patterns of MHW duration, frequency and intensity while just slightly modifying their magnitude.

We also adopt a monthly MHW definition consistent with ref. [56], in which a marine heatwave month is defined as any month during which SST exceeds the 90th percentile of the climatological distribution for that calendar month, based on the full 500-year simulation. The primary motivation for harnessing this monthly definition is to facilitate the analysis of ocean dynamical processes and oceanic heat budget diagnostics, both of which require more variables than SST. Nonetheless, these variables are only provided in the model output at a monthly frequency, not on a daily basis. Compared to MHWs derived from daily SST data (Fig. 1), the monthly MHWs show consistent patterns in duration, annual frequency, and intensity over global oceans (Supplementary Fig. 2), providing a reasonable basis for mechanism-oriented analysis.

In addition to CESM1, we leverage 10 climate models from Coupled Model Intercomparison Project phase 3 (CMIP3, see Supplementary Table 1) that have preindustrial control simulations with either dynamic or slab ocean. We investigate MHWs from these models based on the monthly MHW definition due to data availability. We find that CMIP3 multi-model mean follows generally consistent MHW patterns with CESM1 (Fig. 2 and Supplementary Fig. 2), which demonstrates the robustness and generalizability of our findings.

## El Niño-Southern Oscillation (ENSO)
We compare ENSO variability from CESM1 preindustrial (500 years) and historical (1920-2024) simulations with observations (ERSSTv6, 1920–2024). We discover that CESM1 captures the observed ENSO variability peaking around four years in both preindustrial and

historical simulations, albeit anthropogenic forcings reduce ENSO variability during the historical period (Supplementary Fig. 11). We also compare the observed and simulated extreme ENSO events that are characterized by unusual high SSTs with Niño 3 precipitation exceeding 5 mm/day[30]. Focusing on such events between 1979 and 2024 during which both ERSSTv6 and CMAP data are available, we find that CESM1 historical large ensemble can generally well simulate the observed characteristics during the historical era (Supplementary Fig. 12).

We further examine the Bjerknes (BJ) stability index, which characterizes the growth rate of ENSO by quantifying the strength and mean state of the coupled ocean–atmosphere feedbacks in the equatorial Pacific[57]. The BJ index can be written as

$$BJ = -\alpha_s - \alpha_{MA} + \mu_a\beta_u\langle -\bar{T}_x\rangle + \mu_a\beta_w\langle -\bar{T}_z\rangle + \mu_a\beta_h\left\langle\frac{\bar{w}}{H_1}\right\rangle\alpha_h \quad (1)$$

which is derived from the linearized mixed-layer temperature anomaly equation in the upper 50 m of the tropical east-central Pacific (5°S-5°N, 180°-80°W). The coefficients in the BJ stability index are model dependent and diagnosed directly from the CESM1 simulations. Specifically, the $\alpha$ and $\beta$ parameters evaluate the linear sensitivities of oceanic and atmospheric responses to SST anomalies and are estimated via linear regressions between SST anomalies and the corresponding feedback terms. In Eq. (1), $-\alpha_s$ represents the thermodynamic damping (TD), while $-\alpha_{MA}$ denotes mean advection feedback (MA). The term $\mu_a\beta_u\langle -\bar{T}_x\rangle$ represents the zonal advection feedback (ZA), $\mu_a\beta_w\langle -\bar{T}_z\rangle$ represents the Ekman upwelling feedback (EK) and $\mu_a\beta_h\left\langle\frac{\bar{w}}{H_1}\right\rangle\alpha_h$ represents the thermocline feedback (TH). These terms collectively capture how the background state modulates ENSO growth, including the climatological zonal and vertical temperature gradients and vertical ocean velocity ($\bar{T}_x, \bar{T}_z, \bar{w}$), the sensitivity of wind stress to SST anomalies ($\mu_a$), and the coupling between SST anomalies and anomalous wind stress ($\beta_u, \beta_w,$ and $\beta_h$) that affects thermocline slope, upwelling, and zonal current variations. The parameter $\alpha_h$ represents the impact of subsurface temperature and sea-level anomalies on SST, and $H_1$ denotes an effective depth relevant to vertical advection. In the analysis, we look into the Bjerknes stability for either extreme or normal ENSO events. Based on Niño 3.4 index, we choose extreme ENSO events with SST anomalies exceeding ±2σ and normal ENSO events with SST anomalies of ±1σ to ±2σ, where σ denotes one standard deviation of SST (Supplementary Fig. 3a). This standard-deviation definition of extreme ENSO is consistent with the aforementioned precipitation definition[58].

## Mixed-layer ocean heat budget analysis
To investigate how ocean dynamic processes modulate MHWs, we perform a mixed-layer ocean heat budget analysis for both slab and dynamic oceans. In the form ocean, the tendency of averaged temperate ($\bar{T}$) over the slab, i.e., the ocean mixed layer, is determined by a balance between net surface heat flux and prescribed ocean heat flux convergence[59], as expressed by:

$$\frac{\partial \bar{T}}{\partial t} = \frac{Q_{net} - Q_{flux}}{\rho_0 c_p h} \quad (2)$$

where $Q_{net}$ is net surface heat flux (including shortwave, longwave, latent, and sensible components), $Q_{flux}$ indicates a prescribed ocean heat transport convergence, $\rho_0$ is reference ocean density, $c_p$ is heat capacity of the ocean, and $h$ is mixed layer depth that varies geographically. This simplified framework assumes no active ocean dynamics and is commonly used to separate the thermodynamic response of SST to atmospheric forcing.

In dynamic ocean, the mixed-layer heat budget[60] is derived by vertically integrating from the level of $-h$ to the surface ($z = 0$), following the equation:

$$\rho_0 c_p \int_{-h}^{0} \frac{\partial T}{\partial t} dz = \rho_0 c_p \int_{-h}^{0} (-\nabla \cdot \boldsymbol{u}T + HMIX + P)dz + Q_{net} - \kappa \left(\frac{\partial T}{\partial z} - \Gamma\right)_{z=-h} - Q_{p,-h} \tag{3}$$

where $T$ is ocean temperature in the mixed layer, $\nabla$ denotes 3D gradient operator, $\boldsymbol{u}$ denotes 3D ocean velocity, $HMIX$ denotes horizontal mixing, $P$ denotes the convergence of parameterized transport, $\kappa$ is vertical diffusivity, $\Gamma$ is the KPP countergradient flux of temperature and $Q_{p,-h}$ is the penetrative heat flux at the level of $-h$. Thus, the tendency of averaged temperate over the mixed layer in dynamic ocean is

$$\frac{\partial \bar{T}}{\partial t} = \frac{1}{h} \int_{-h}^{0} (-\nabla \cdot \boldsymbol{u}T + HMIX + P)\partial z + \frac{Q}{\rho_0 c_p h} - \frac{\kappa}{\rho_0 c_p h}\left(\frac{\partial T}{\partial z} - \Gamma\right)_{z=-h} - \frac{Q_{p,-h}}{\rho_0 c_p h} \tag{4}$$

where $\frac{\partial \bar{T}}{\partial t} = \frac{1}{h}\int_{-h}^{0}\frac{\partial T}{\partial t}dz$. Through the heat budget analysis, we are able to quantify the contributions of surface heat fluxes, ocean advection, horizontal and vertical mixing to mixed-layer temperature tendency to determine the dominant factor on MHW evolution.

## Potential predictability variance fraction analysis
To detect the predictability of MHW duration, annual frequency, and intensity, we calculate the potential predictability variance fraction (ppvf) using the methods of refs. 34,35. The ppvf is defined as a fraction of long-time scale (or low frequency) variability $\sigma_L^2$ with respect to the total variability $\sigma^2$. The total climate variability can be decomposed into a low-frequency component $\sigma_L^2$ and a residual noise component $\sigma_\varepsilon^2$, such that:

$$ppvf = \frac{\sigma_L^2}{\sigma^2} = \frac{\sigma_L^2}{\sigma_\varepsilon^2 + \sigma_L^2} \tag{5}$$

Here, $\sigma_L^2$ is estimated as the variance of MHW duration, annual frequency, or intensity after applying an $x$-year running mean (e.g., 5-year, 10-year, or 25-year), which filters out high-frequency fluctuations. The high ppvf regions identify those in which long timescale variability stands out clearly from short timescale variability, and thus variability in those regions may be at least potentially predictable.

## Average predictability time analysis
To quantify the dominant predictable components of MHW duration, annual frequency, and intensity, we apply the diagnostic Average Predictability Time (APT) method that was developed by refs. 61,62, and broadly used in climate predictability studies[63,64]. APT provides an objective measure of how long a given mode of variability remains predictable above climatological noise, which is defined as the integral of predictability over all the lead times:

$$APT = 2\sum_{\tau=1}^{\infty} \left(\frac{\sigma_\infty^2 - \sigma_\tau^2}{\sigma_\tau^2}\right) = 2\sum_{\tau=1}^{\infty}\left(\frac{\boldsymbol{q}^T\left(\sum_\infty \boldsymbol{q} - \sum_\tau \boldsymbol{q}\right)}{\boldsymbol{q}^T\sum_\infty \boldsymbol{q}}\right) \tag{6}$$

where $\sigma_\infty^2$ denote climatological variance, $\sigma_\tau^2$ denote the ensemble forecast variance at a lead time $\tau$ and $\mathbf{q}$ is a projection vector. This measure approaches one for a perfect forecast and zero when the ensemble forecast spread approaches the climatological spread. Since the preindustrial simulation used in this study only has one ensemble member, we estimate APT using a linear regression-based framework rather than ensemble-based forecast error covariance. The regression

model is:

$$\hat{\boldsymbol{x}}_{t+\tau} = \boldsymbol{L}_\tau \boldsymbol{x}(t) + \boldsymbol{\epsilon}(t) \tag{7}$$

where $x(t)$ is the predictor at time $t$, $\hat{x}_{t+\tau}$ is the predictand at time $t + \tau$, $\boldsymbol{L}_\tau$ is the regression coefficient and $\boldsymbol{\epsilon}(t)$ is a residual term. Then, the climatology and forecast matrices have the following form:

$$\sum_\infty = \boldsymbol{C}_0$$
$$\sum_\tau = \boldsymbol{C}_0 - \boldsymbol{C}_\tau \boldsymbol{C}_0^{-1}\boldsymbol{C}_\tau^T \tag{8}$$

where $C_0$ is the climatological variance, $C_\tau$ denotes the time lagged covariance matrix.

Combining Eqs. (5)–(7), the eigenvalue problem caused by maximization of APT becomes:

$$\left(2\sum_{\tau=1}^{\infty}\boldsymbol{C}_\tau \boldsymbol{C}_0^{-1}\boldsymbol{C}_\tau^T\right)\boldsymbol{q} = \lambda \boldsymbol{C}_0 \boldsymbol{q} \tag{9}$$

where $\lambda$ denote the eigenvalues which are the APT values corresponding to each component. The left side in Eq. (9) represents the integration of signal covariance, and the right side represents the total climatological covariance.

We project the predictor variables and the dependent variables onto the first 30 principal components (PCs), which results in a 500-year PC time series for CESM1 preindustrial simulations. The PCs are then divided into two parts: the first 250 years are used as training data to estimate the optimal APT component, and the remaining 250 years are used for verification[65]. Following ref. 62, we adopt the squared multiple correlation $R_\tau^2$ to estimate the potential predictability at each lead time $\tau$.

$$R_\tau^2 = \frac{\mathbf{q}^T \mathbf{C}_\tau \mathbf{C}_0^{-1}\mathbf{C}_\mathbf{q}^T}{\mathbf{q}^T \mathbf{C}_0 \mathbf{q}} \tag{10}$$

where $\boldsymbol{q}$ is calculated from the training data, and the covariance terms are obtained from verification data. The rate by which $R_\tau^2$ declines over time reflects the predictability of each component. The slower the decline in $R_\tau^2$, the higher the potential predictability.

Aside from CESM1 preindustrial simulations, we apply the APT method for CESM1 historical large ensemble from 1920 to 2024. Benefited from large ensemble (40 members), the ensemble mean represents climate response to anthropogenic forcing. We remove the ensemble mean from each member, leaving only the residual, which represents the internal variability as in line with the preindustrial control. We repeat our analysis on APT1 for each member's residual and find the result is largely similar to the preindustrial control (Supplementary Fig. 9). This supports the methodological reliability, indicating that our MHW predictability persists under anthropogenic forcing.

## Significance test
We use a two-sample Student's t test to determine the significance of the differences in MHW duration, annual frequency, and intensity between dynamic and slab ocean experiments. We conduct the test at each grid point to see whether the differences are statistically significant at the 95% confidence level.

We also assess the statistical significance of the APT through Monte Carlo experiments. We generate two independent random matrices with zero mean and unit variance and put them into Eq. (9) to obtain a set of optimized ordered sequence of APT values. We repeat this procedure 100 times to construct a reference distribution of eigenvalues under the null hypothesis. Then, we select the 95th percentile of this distribution as the significance threshold. If the APT

value computed from the training data exceeds this threshold, we reject the null hypothesis and conclude that the APT value is statistically significant at the 5% level.

In addition, we assess the statistical significance of the spatial correlation patterns between APT1 and the AMOC stream-function using a two-tailed significance test based on the 95% confidence threshold.

## Data availability

CESM1 model data are available at https://gdex.ucar.edu/datasets/d651027/dataaccess/. CMIP3 model data are available at https://aims2.llnl.gov/search. ERSSTv5 data are available at https://www.ncei.noaa.gov/products/extended-reconstructed-sst. CMAP Precipitation data are available at https://psl.noaa.gov/data/gridded/data.cmap.html.

## Code availability

Figures are generated via the NCAR Command Language (NCL, Version 6.5.0) [Software]. (2018). Boulder, Colorado: UCAR/NCAR/CISL/TDD (https://doi.org/10.5065/D6WD3XH5). The codes to generate Figs. 1–5 are available from Zenodo[66] at https://doi.org/10.5281/zenodo.18357158.

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

## Acknowledgements

This study has been supported by U.S. National Science Foundation (OCE-2123422, AGS-2053121, and AGS-2237743). We thank Timothy M. DelSole for helpful comments and suggestions.

## Author contributions

X.R. performed the analysis and wrote the original draft of the paper. W.L. conceived the study. L.Z. guided the predictability analysis. All authors contributed to interpreting the results and made improvements to the paper.

## Competing interests

The authors declare no competing interests.
