## [Transparent Peer Review file · Nature Communications]

Ocean dynamics shape marine heatwaves and their predictability

Corresponding Author: Dr Wei Liu

Version 0:

Reviewer comments:

Reviewer #1

(Remarks to the Author)

The manuscript entitled “Ocean dynamics shape marine heatwaves and their predictability” aims to investigate the role of ocean dynamics in the evolution and predictability of marine heatwaves (MHWs) at the global scale through a comparison of slab ocean and fully coupled simulations. The research topic is important, and the overall writing is clear. There are a few points which need to be clarified.

Major comments :

1. The authors claim that “the contribution of ocean dynamics to MHW evolution on a global scale has yet to be systematically evaluated.” However, previous studies have already examined the link between MHWs and ENSO at the global scale. For example, Gregory et al. (2024, GRL) conducted a systematic analysis of the impact of different types of ENSO on global MHWs, which overlaps partially with the discussion of ENSO–MHW in this manuscript. In addition, the modulation of MHWs by the AMOC has been clearly discussed in their previous study (Ren & Liu, 2021, GRL), and the current manuscript seems to provide a limited extension of this work. Overall, the claim of a “first systematic evaluation” is a bit difficult to justify, and the manuscript should more explicitly highlight its differences and novel contributions relative to prior studies.
2. The study relies solely on a single model (CESM1) pre-industrial long-term simulation, lacking multi-model comparison or validation with observational or reanalysis data. This limits the robustness and generalizability of the results. Moreover, the study is based on idealized control experiments and does not consider the evolution of MHWs under greenhouse gas forcing, reducing its relevance to real-world climate risk and applications. Additionally, the APT analysis is conducted using a single ensemble member’s long-term time series, which may suffer from overfitting or insufficient statistical significance, warranting further evaluation of the methodological reliability.
3. The ENSO-related regional analysis is mostly qualitative, lacking quantitative assessment of the underlying physical mechanisms. For the North Atlantic, although the role of the AMOC is emphasized, the results largely overlap with previous studies, appearing more as a confirmation rather than a new scientific insight. While the manuscript frequently emphasizes “global-scale conclusions,” the results are mostly limited to several typical regions, reducing their broader applicability.
4. The manuscript emphasizes the modulation of MHWs by ENSO-related regions and the AMOC, but the predictability analysis mainly relies on statistical metrics such as potential predictability fraction (ppvf) and APT. Interpreting these metrics directly as “actual predictive skill” risks conflating statistical potential with real-world forecasting ability.

Ref :

Gregory, C. H., Artana, C., Lama, S., León-FonFay, D., Sala, J., Xiao, F., et al. (2024). Global marine heatwaves under different flavors of ENSO. *Geophysical Research Letters*, 51, e2024GL110399. <https://doi.org/10.1029/2024GL110399>

Ren, X., & Liu, W. (2021). The role of a weakened Atlantic meridional overturning circulation in modulating marine heatwaves in a warming climate. *Geophysical Research Letters*, 48, e2021GL095941. <https://doi.org/10.1029/2021GL095941>

Minor comments

1. For Figure 1: It is recommended to clearly label the different maps with "Dynamic minus Slab" in the legend to avoid any ambiguity. Currently, relying solely on the caption for explanation is not sufficiently intuitive.
2. Line 104-111 : The sentence “existence of ocean dynamical process also diminish the intensify of MHWs” is inaccurate. It should read “the existence of ocean dynamical processes reduces the intensity of MHWs” for both grammatical correctness

and scientific clarity.

3. Line 122-137: The analysis appears to be based only on monthly composites (month 0 ± 6). It is unclear whether this approach can adequately capture short-lived intense events or interannual variability.

4. Line 166-175: The "wind-evaporation-SST (WES) feedback" explanation sounds reasonable, but the manuscript does not provide quantitative evidence or references supporting the role of WES or cloud-radiative feedback. The authors should either explicitly analyze these processes or cite relevant literature to substantiate this claim.

5. Line 236-248: Interpreting the differences between slab and dynamic ocean simulations as the "contribution of ocean dynamics" is a reasonable approach. However, these differences also include variations in air-sea coupling feedbacks, not just ocean dynamics per se. The authors should be more cautious in phrasing this throughout the manuscript and explicitly acknowledge this point.

Reviewer #2

(Remarks to the Author)

This is my review of the paper "Ocean dynamics shape marine heatwaves and their predictability" submitted to Nature Communications.

The paper analyzes the sensitivity of Marine heatwaves to the ocean dynamics by examining a fully coupled climate model against a slab ocean model, in which ocean dynamics is not present. The analysis of marine heatwave metrics shows five regions in which these parameters change between the two ensemble simulations. Two main regions are explored: i) the east tropical Pacific, where El Niño variability dominates over interannual periods and the vertical mixing and advection are important for the sea surface temperature modulation, and ii) the subpolar North Atlantic, where the AMOC is known to affect the SST resulting in a dipole across the Gulf Stream extension. Predictability increases in the north Atlantic region due to the AMOC, and reduces in the Niño 3.4 region due to the anomaly advection away from the region.

The paper is well written and presented. The results in the two main regions are expected, since the two main regions explored are well explained dynamically in numerous past studies. The idea of using a slab ocean to analyze these differences and the effect on the SST response during MHWs is novel in my view, so the results are important for this quantification.

I have four main comments, which include comparison to observations, the seasonality of mixed layer variability in the slab ocean, the location of the formation of anomalies of the AMOC into the subpolar North Atlantic, and the effect of ice formation anomaly to the North Atlantic in addition to the AMOC. Also included are some minor additional comments. Apart from those, I think this paper is relevant for the community, and could be suitable for publication in Nature Communications after revision.

Main comment:

i) Some comparison with observations would be welcome. For example, figures 1 and 2 could easily be drawn from observations for a comparison. This is important to elucidate some comments such as in L. 90-91 that "ocean heat transport and vertical mixing are more realistically represented in dynamic ocean". At least some comparison to observational studies in terms of SST variability and precipitation anomalies should be included.

ii) The slab ocean does not have a seasonal mixed layer variability, just a spatially varying annual mean. The seasonality of the mixed layer is a very important feature to the formation of water masses in the ocean, and in particular in the analysis of the subpolar North Atlantic region. By suppressing it, there would probably be an enhancement of the SST variability for the lack of compensating effects in the mixed layer. This hasn't been mentioned in the paper, and the caveats and associated effects should be referred to and their caveats included in the paper.

iii) The AMOC variability related to the APT1 mode: The AMOC variability in interannual to decadal timescales should be related to anomalies that occur in the subpolar regions, and then propagated southward (e.g., Petit et al., 2025 <https://doi.org/10.1029/2025GL115171>) before influencing the region. This could come from either heat flux or salinity anomalies in the subpolar north Atlantic. Figure 5 shows a simple correlation. Maybe evolution maps could be shown to pinpoint the source of the anomalies.

iv) For completeness, in addition to the effect of the AMOC, sea ice is also an important forcing in the subpolar gyre warming/cooling. Can this effect be quantified in the model? This is one of the main uncertainties related to the AMOC spread in models, and since only one model was used here, it would be good to add this perspective. This analysis could be supplementary or in consonance with the previous point. Adding this analysis would considerably improve the knowledge of the effects of the ocean to the MHWs in the North Atlantic.

Minor comments:

L.33 "eastern tropical Pacific" for consistency throughout the manuscript.

L. 99 "we see more frequent extreme ENSO episodes in dynamic ocean". Figure 1f seems to show otherwise, with the SST anomalies in the ENSO region being less frequent in the dynamic ocean. I agree with L. 102 that they are more intense and persistent.

L. 131-133 easterly winds do not only produce westward advection, but also equatorial divergence and increase of vertical upwelling. This should be noted.

L. 188 typo which "could" be understood.

L. 193 Does this "positive anomalies" refer to a different phase of the variability or of a particular region in the map?

Version 1:

Reviewer comments:

Reviewer #1

(Remarks to the Author)

The authors have responded to my review comments. Their key findings are that ocean dynamics can significantly enhance the intensity and prolong the duration of marine heatwaves in mid-to-high latitude oceans, and that marine heatwaves exhibit robust multi-year potential predictability. This is an important insight for improving heatwave prediction, given the frequent occurrence of such events. I recommend the acceptance of the manuscript.

Reviewer #2

(Remarks to the Author)

This is my review of the revised version of the paper "Ocean dynamics shape marine heatwaves and their predictability". The authors addressed all my comments, including comparison with observations and acknowledging the caveats associated with this work. I have a few minor additional comments that I would like the authors to address before my final acceptance.

Minor Comments:

L. 73: Has yet "to be" studied

L. 88 ENSO power "spectrum"

L. 126 We "compose the heat budget" or "make a composite of heat budget" not "we composite the heat budgets"

L 228-231 - It is worth mentioning that ocean dynamics reduces the predictability of the ENSO region on interannual-to-decadal timescales.

L. 338-352 The Bjerknes feedback terms contain several constants (alphas and betas). As I understand, these should be model dependent, representing for example the sensitivity of the ocean to wind forcing, and should be derived from these relationships. How were they derived? Please explain in the text.

Reviewer #1 (Remarks to the Author):

The manuscript entitled “Ocean dynamics shape marine heatwaves and their predictability” aims to investigate the role of ocean dynamics in the evolution and predictability of marine heatwaves (MHWs) at the global scale through a comparison of slab ocean and fully coupled simulations. The research topic is important, and the overall writing is clear. There are a few points which need to be clarified.

Thanks for the insightful advice and comments!

Major comments:

1. The authors claim that “the contribution of ocean dynamics to MHW evolution on a global scale has yet to be systematically evaluated.” However, previous studies have already examined the link between MHWs and ENSO at the global scale. For example, Gregory et al. (2024, GRL) conducted a systematic analysis of the impact of different types of ENSO on global MHWs, which overlaps partially with the discussion of ENSO–MHW in this manuscript. In addition, the modulation of MHWs by the AMOC has been clearly discussed in their previous study (Ren & Liu, 2021, GRL), and the current manuscript seems to provide a limited extension of this work. Overall, the claim of a “first systematic evaluation” is a bit difficult to justify, and the manuscript should more explicitly highlight its differences and novel contributions relative to prior studies.

Thanks for the providing the reference (Gregory et al. 2024, GRL). We agree and tone down as “Despite different types of ENSO events were suggested to affect global MHWs via teleconnection in which dynamical ocean processes could be involved (Gregory et al. 2024), the contribution of ocean dynamics to MHW evolution on a global scale still needs to be explicitly and systematically evaluated. More importantly, the associated MHW predictability has yet been studied over a global scale.”.

Also, thanks for mentioning our earlier study (Ren and Liu 2021, GRL, RL21 thereafter). RL21 is different from the current study from two aspects. i) RL21 examined AMOC impacts on MHWs under climate change by comparing the climatology states with free and fixed AMOCs. Each climatology is represented by a 40-year average (1981-2020 or 2061-2100). The current study, on the other hand, probed the role of AMOC (inter-)decadal variability (mainly internal variability, not its centurial decline due to anthropogenic warming) on MHWs and their predictability. We focused on the lead/lag correlation between AMOC and MHW variations since this is the key for predictability, whereas such several-year lead/lag does not much matter for a 40-year average as in RL21. ii) Spatial patterns are different. We showed a dipole-like pattern in the most predictable component (APT1) for MHW frequency (Fig. 5a in the original submission), meaning that a decelerated AMOC will cause negative anomalies (less frequent MHWs) in the Labrador Sea/South Greenland while positive anomalies (more frequent MHWs) in the Gulf Stream region. Nevertheless, the latter positive anomalies (more frequent MHWs by an AMOC decline) in the Gulf Stream region did not significantly exhibit in RL21 over 1981-2020. Moreover, in RL21 over 2061-2100, the AMOC decline leads to more frequent MHWs in the Labrador Sea/South Greenland, in that the region will be in a near-permanent MHW state without an AMOC slowdown. This result differs from the current finding.

To summarize, RL21 and the current study reveal the AMOC influence on MHWs from two different aspects, i.e., climate change, and natural decadal to multidecadal variability.

2. The study relies solely on a single model (CESM1) pre-industrial long-term simulation, lacking multi-model comparison or validation with observational or reanalysis data. This limits the robustness and generalizability of the results. Moreover, the study is based on idealized control experiments and does not consider the evolution of MHWs under greenhouse gas forcing, reducing its relevance to real-world climate risk and applications. Additionally, the APT analysis is conducted using a single ensemble member's long-term time series, which may suffer from overfitting or insufficient statistical significance, warranting further evaluation of the methodological reliability.

Thank you so much for your valuable comments! In response to the reviewer, we compare CESM1 preindustrial MHWs with observations (OISST v2.1, 1982-2024), and also with CESM1 historical MHWs during the same period (1982-2024). We find consistent MHW characteristics amongst the three suites over a global scale, such as high-intensity MHWs of long duration mainly in the eastern tropical Pacific, and short-lived, high-intensity MHWs in the Gulf Stream and Kuroshio extension regions (Fig. R1). Compared with observations, CESM1 historical and preindustrial runs simulate an overall longer MHW duration, a lower annual mean MHW frequency and a weaker MHW intensity, which are common issues in MHW simulations by non-eddy-resolving models (Pilo et al. 2019). We conjecture that the discrepancy between model and observation might come from the fact that in observation the daily SST climatology is based on this sole realization. Whilst in model a 40-member ensemble mean (for historical) or a 500-year average (for preindustrial) daily SST climatology is used for MHW detections in different ensemble members so that some short-duration MHWs could be hard to be determined or potentially mixed with intense long-duration MHWs. Comparisons between CESM1 historical and preindustrial simulation further evince that anthropogenic aerosol and greenhouse gas forcing do not alter the global patterns of MHW duration, frequency and intensity while just slightly modifying their magnitude. All these aforementioned discrepancies, however, do not affect the conclusion of our study. We have included the above discussions and Fig. R1 in the revision.

Fig. R1: (a-c) MHW durations of (a) OISST v2.1 and (b) CESM1 historical ensemble mean over 1982-2024, as well as (c) CESM1 preindustrial simulations. (d-f) Same as (a-c) but for annual mean MHW frequencies. (g-i) Same as (a-c) but for MHW intensity.

Model	Dynamic ocean (years)	Slab ocean (years)
CGCM3.1(T47)	1001	30
CGCM3.1(T63)	1001	30
CSIRO-Mk3.0	380	60
GFDL-CM2.0	500	50
GFDL-CM2.1	500	100
GISS-ER	500	120
INM-CM3.0	330	60
MIROC3.2 (medres)	500	60
ECHAM5/MPI-OM	506	100
UKMO-HadCM1	240	71

Table R1. CMIP3 models used in this study, and their experiment durations for simulations with dynamic and slab oceans.

We have also compared the fully coupled and slab-ocean preindustrial simulations with multiple CMIP3 models (Table R1). We use the monthly MHW definition as only monthly SSTs are available. CMIP3 multi-model mean (Fig. R2) shows generally consistent patterns with CESM1, which demonstrates the robustness and generalizability of our results. We have included the above discussion as well as Table R1 and Fig. R2 in the revision.

Fig. R2: (a-c) MHW durations (color shading in months) of (a) dynamic and (b) slab oceans, as well as (c) the difference between the two (dynamic minus slab) for the multi-model mean of CMIP3 preindustrial simulations. (d-f) Same as (a-c) but for MHW annual frequencies. (g-i) Same as (a-c) but for MHW intensities. Different from Fig. 1, MHWs here are based on the monthly MHW definition (Methods). The stipples in panels (c, f, i) refer to the regions where differences are not statistically insignificant based on Student's t-test at 95%

confidence level.

Fig. R3: (a) The leading predictable component (APT1) of North Atlantic MHW annual frequency in CESM1 historical simulations (Methods), and (b) the squared multiple correlation coefficients R^2 . The dashed black line denotes the 95% significance level. (c) Lead/lag correlation between Atlantic meridional overturning stream-function and APT1 index for North Atlantic MHW annual frequency. Lags range from -14 to -2 years, in which the negative lags indicate that the AMOC leads the APT index. The stipples refer to the regions where correlations are statistically insignificant at 95% confidence level.

To further evaluate the methodological reliability, we make use of CESM1 historical large ensemble simulations from 1920 to 2024. Benefited from the large ensemble (40 members), the ensemble mean represents the climate response to anthropogenic forcing. We thus remove the ensemble mean from each member such that the residual represents the internal variability, in line with the preindustrial control. We repeat our analysis on APT1 for the residual of each member (Fig. R3) and find that the result is consistent with that from the preindustrial control. This justifies the methodological reliability, meaning that our MHW predictability persists under anthropogenic forcing. We have included the above discussion and Fig. R3 in the revision.

Pilo, G. S. et al. Sensitivity of marine heatwave metrics to ocean model resolution. *Geophys. Res. Lett.* **46**, 14604–14612 (2019).

3. The ENSO-related regional analysis is mostly qualitative, lacking quantitative assessment of the underlying physical mechanisms. For the North Atlantic, although the role of the AMOC is emphasized, the results largely overlap with previous studies, appearing more as a confirmation rather than a new scientific insight. While the manuscript frequently emphasizes “global-scale conclusions,” the results are mostly limited to several typical regions, reducing their broader applicability.

Please see our previous reply (response to comment 1) to clarify the difference on AMOC effect between previous studies and our current study.

In response to the comment, we further examine the underlying physical mechanisms on extreme ENSO to better understand the MHWs in this region. Based on Niño 3.4 index, we choose extreme ENSO events with SST anomalies exceeding $\pm 2\sigma$ and normal ENSO events with SST anomalies of $\pm 1\sigma$ to $\pm 2\sigma$, where σ denotes SST standard deviation (Fig. R4a). We look into the Bjerknes (BJ) stability index, which depends on the thermodynamic damping (TD), mean advection feedback (MA), zonal advection feedback (ZA), Ekman upwelling feedback (EK) and thermocline feedback (TH). These feedbacks regulate the linear stability of the ENSO system and thus influence its amplitude and development. We find that, although extreme ENSO exhibits larger magnitude than normal ENSO in all feedback terms, its thermocline feedback has the most positive contribution to the BJ index and is the primary factor to make its BJ index higher than that of normal ENSO (Fig. R4b). We have included the above discussion and Fig. R4 in the revision.

Fig. R4: (a) Niño 3.4 index from the dynamic ocean simulation, in which extreme ENSO

events are defined as SST anomalies exceeding $\pm 2\sigma$ (beyond the red and blue lines), and normal ENSO events are defined as SST anomalies from $\pm 1\sigma$ to $\pm 2\sigma$ (between the orange and light blue lines) where σ denotes one standard deviation of SST. **(b)** The Bjerkness (BJ) index and individual components (Methods) of normal ENSO (orange) and extreme ENSO (red). TD, MA, ZA, EK, and TH represent the thermal damping, mean advection, zonal advection, Ekman upwelling, and thermocline feedbacks, respectively. Error bars denote one standard deviation among ENSO events.

4. The manuscript emphasizes the modulation of MHWs by ENSO-related regions and the AMOC, but the predictability analysis mainly relies on statistical metrics such as potential predictability fraction (ppvf) and APT. Interpreting these metrics directly as “actual predictive skill” risks conflating statistical potential with real-world forecasting ability.

Both ppvf and APT of CESM1 preindustrial control runs detect the perfect model skill, which assumes that we have perfect initiation conditions and there are no model biases. Thus, this is the upper limit of the prediction skill and we usually called “predictability” rather than real prediction skill. As mentioned by the reviewer, in reality, the prediction skill is usually smaller than this perfect model skill because we have model bias, and the initial conditions are not perfect. We have included the above discussion in the revision.

Ref:

Gregory, C. H., Artana, C., Lama, S., León-FonFay, D., Sala, J., Xiao, F., et al. (2024). Global marine heatwaves under different flavors of ENSO. *Geophysical Research Letters*, 51, e2024GL110399. <https://doi.org/10.1029/2024GL110399>

Ren, X., & Liu, W. (2021). The role of a weakened Atlantic meridional overturning circulation in modulating marine heatwaves in a warming climate. *Geophysical Research Letters*, 48, e2021GL095941. <https://doi.org/10.1029/2021GL095941>

Minor comments

1. For Figure 1: It is recommended to clearly label the different maps with "Dynamic minus Slab" in the legend to avoid any ambiguity. Currently, relying solely on the caption for explanation is not sufficiently intuitive.

Revised as suggested.

2. Line 104-111: The sentence “existence of ocean dynamical process also diminish the intensify of MHWs” is inaccurate. It should read “the existence of ocean dynamical processes reduces the intensity of MHWs” for both grammatical correctness and scientific clarity.

Revised as suggested.

3. Line 122-137: The analysis appears to be based only on monthly composites (month 0 ± 6). It is unclear whether this approach can adequately capture short-lived intense events or interannual variability.

MHWs across the Niño region exhibit distinct characteristics in maps based on both daily and monthly MHW definitions, which supports our use of monthly data for future study as an approximation to daily data. On the other hand, since the model did not output the variables required for heat budget analysis on a daily basis, we can only perform monthly heat budget analysis, which, as the reviewer mentioned, may not well capture MHWs over sub-month timescales. Here, our analysis based on monthly composites aims to characterize

the mean evolution of MHW-related heat budget tendencies. While monthly averaging may smooth the most transient anomalies, most MHWs included in our analysis persist for multiple weeks to months, and their key thermodynamic signatures remain robust at a monthly resolution.

4. Line 166-175: The “wind-evaporation-SST (WES) feedback” explanation sounds reasonable, but the manuscript does not provide quantitative evidence or references supporting the role of WES or cloud-radiative feedback. The authors should either explicitly analyze these processes or cite relevant literature to substantiate this claim.

Following the review, we have added the two references (Espinosa et al. 2024; Middlemas et al. 2019) to support the discussion of WES and cloud-radiative feedbacks.

Espinosa, W. et al. The shortwave cloud-SST feedback amplifies multidecadal variability in the southeast Pacific. *Geophys. Res. Lett.* **51**, e2024GL111039 (2024).

Middlemas, M. S., Zhang, H. & Clement, A. C. Contributions of atmospheric and oceanic feedbacks to northeast subtropical SST variability. *Clim. Dyn.* **52**, 2371–2388 (2019).

5. Line 236-248: Interpreting the differences between slab and dynamic ocean simulations as the “contribution of ocean dynamics” is a reasonable approach. However, these differences also include variations in air–sea coupling feedbacks, not just ocean dynamics per se. The authors should be more cautious in phrasing this throughout the manuscript and explicitly acknowledge this point.

Following the review, we have revised the text as “Comparing the dynamic and slab ocean simulations provides insight into how interactive ocean dynamics, along with coupled feedback processes, influence the evolution and predictability of global marine heatwaves (MHWs).”.

Reviewer #2 (Remarks to the Author):

This is my review of the paper "Ocean dynamics shape marine heatwaves and their predictability" submitted to Nature Communications.

The paper analyzes the sensitivity of Marine heatwaves to the ocean dynamics by examining a fully coupled climate model against a slab ocean model, in which ocean dynamics is not present. The analysis of marine heatwave metrics shows five regions in which these parameters change between the two ensemble simulations. Two main regions are explored: i) the east tropical Pacific, where El Nino variability dominates over interannual periods and the vertical mixing and advection are important for the sea surface temperature modulation, and ii) the subpolar North Atlantic, where the AMOC is known to affect the SST resulting in a dipole across the Gulf Stream extension. Predictability increases in the north Atlantic region due to the AMOC, and reduces in the Nino 3.4 region due to the anomaly advection away from the region.

The paper is well written and presented. The results in the two main regions are expected, since the two main regions explored are well explained dynamically in numerous past studies. The idea of using a slab ocean to analyze these differences and the effect on the SST response during MHWs is novel in my view, so the results are important for this quantification.

I have four main comments, which include comparison to observations, the seasonality of mixed layer variability in the slab ocean, the location of the formation of anomalies of the AMOC into the subpolar North Atlantic, and the effect of ice formation anomaly to the North Atlantic in addition to the AMOC. Also included are some minor additional comments. Apart from those, I think this paper is relevant for the community, and could be suitable for publication in Nature Communications after revision.

Thanks for the insightful advice and comments!

Main comment:

i) Some comparison with observations would be welcome. For example, figures 1 and 2 could easily be drawn from observations for a comparison. This is important to elucidate some comments such as in L. 90-91 that "ocean heat transport and vertical mixing are more realistically represented in dynamic ocean". At least some comparison to observational studies in terms of SST variability and precipitation anomalies should be included.

Thank you so much for your valuable suggestions! Following the review, also as Reviewer 1 suggested, we have compared CESM1 preindustrial MHWs with observations (OISST v2.1, 1982-2024), and also with CESM1 historical MHWs during the same period (1982-2024). We find consistent MHW characteristics amongst the three suites over a global scale, such as high-intensity MHWs of long duration mainly in the eastern tropical Pacific, and short-lived, high-intensity MHWs in the Gulf Stream and Kuroshio extension regions (Fig. R1). Compared with observations, CESM1 historical and pre-industrial runs simulate an

overall longer MHW duration, a lower annual mean MHW frequency and a weaker MHW intensity, which is a common issue in MHW simulations by non-eddy-resolving models (Pilo et al. 2019). We conjecture that the discrepancy between model and observation comes from the fact that in observation the daily SST climatology is based on this sole realization. While in model a 40-member ensemble mean (for historical) or a 500-year average (for preindustrial) daily SST climatology is used for MHW detections in different ensemble members so that some short-duration MHWs are difficult to be determined or potentially mixed with intense long-duration MHWs. In addition, comparisons between CESM1 historical and preindustrial simulation evince that anthropogenic aerosol and greenhouse gas forcing do not alter the global patterns of MHW duration, frequency and intensity while just slightly modifying their magnitude. All these aforementioned discrepancies, however, do not affect the conclusion of our study.

We have included the above discussions and Fig. R1 in the revision.

Fig. R1: (a-c) MHW durations of (a) OISST v2.1 and (b) CESM1 historical ensemble mean over 1982-2024, as well as (c) CESM1 preindustrial simulations. (d-f) Same as (a-c) but for annual mean MHW frequencies. (g-i) Same as (a-c) but for MHW intensity.

Pilo, G. S. et al. Sensitivity of marine heatwave metrics to ocean model resolution. *Geophys. Res. Lett.* **46**, 14604–14612 (2019).

Following the review, we compare ENSO variability from CESM1 pre-industrial (500 years) and historical (1920-2024) simulations with observations (ERSSTv6, 1920-2024). We discover that CESM1 captures the observed ENSO variability peaking around four years in both preindustrial and historical simulations, albeit anthropogenic forcings reduce ENSO variability during the historical period (Fig. R5). We also compare the observed and simulated extreme ENSO events that are characterized by unusual high SSTs with Niño 3 precipitation exceeding 5 mm/day. Focusing on such events between 1979 and 2024 during which both ERSSTv6 and CMAP data are available, we find that CESM1 historical large ensemble can generally well simulate the observed characteristics during the historical era (Fig. R6).

We have included the above discussion as well as Fig. R5 and Fig. R6 in the revision.

Fig. R5: Power spectra of the Niño 3.4 indices from ERSSTv6 (black) and CESM1 historical large ensemble simulations during 1920-2024 (red, ensemble mean; light red, one standard derivation among ensembles), and CESM1 preindustrial simulation (blue), as well as their 95% confidence limits (dashed/dotted curves).

Fig. R6: Scatterplots of monthly SST versus monthly precipitation in the Niño 3 region for (a) CESM1 historical large ensemble simulations and (b) ERSSTv6 and CMAP observations. Orange dots represent cases when precipitation is larger than 5 mm/day. The black vertical and horizontal dashed lines denote the climatological mean values of SST and precipitation, respectively.

ii) The slab ocean does not have a seasonal mixed layer variability, just a spatially varying annual mean. The seasonality of the mixed layer is a very important feature to the formation of water masses in the ocean, and in particular in the analysis of the subpolar North Atlantic region. By suppressing it, there would probably be an enhancement of the SST variability for the lack of compensating effects in the mixed layer. This hasn't been mentioned in the paper, and the caveats and associated effects should be referred to and their caveats included in the paper.

Thank you very much for your great comments and we totally agree with them. Yes, the slab ocean model adopts annual mean mixed layer depth throughout the year, and does not account for seasonal mixed layer variability, whereas MHWs could be associated with mixed layer depth variations (Sun et al. 2024). We have included the above discussion in the revision.

Sun, W., Wang, Y., Yang, Y., Yang, J., Ji, J., & Dong, C. Marine heatwaves/cold-spells associated with mixed layer depth variation globally. *Geophys. Res. Lett.* **51**, e2024GL112325 (2024).

iii) The AMOC variability related to the APT1 mode: The AMOC variability in interannual to decadal timescales should be related to anomalies that occur in the subpolar regions, and then propagated southward (e.g., Petit et al., 2025 <https://doi.org/10.1029/2025GL115171>) before influencing the region. This could come from either heat flux or salinity anomalies in the subpolar north Atlantic. Figure 5 shows a simple correlation. Maybe evolution maps could be shown to pinpoint the source of the anomalies.

We agree with the reviewer that the buildup the dipole pattern is related to the southward propagation of the AMOC and associated meridional heat transport anomalies from the North Atlantic deep water formation region, at the speed of slow tracer advection along the interior pathways of the deep water (Zhang & Zhang 2015). As the reviewer pointed out, pinpointing the source of the anomalies necessitates an understanding of AMOC variability, since over a cycle, from a positive to negative and back to positive phase of AMOC variability, all variables such as ocean temperatures, salinity, AMOC strength and induced heat and freshwater transports, surface heat and freshwater fluxes as well as winds are looped in and feedback each other. As a result, understanding AMOC variability necessitates an understanding of the entire loop, i.e., how various factors interact to complete a cycle.

[FIGURE REDACTED]

Fig. R7: Power spectrum for AMOC magnitude at 45°N in the CESM1 preindustrial simulation. Dash lines denote the 95 % confidence level.

AMOC variability was suggested as a damped ocean mode excited by stochastic atmospheric forcing (Delworth et al. 1993; Griffies & Tziperman 1995). In particular, AMOC variability with a 20–30 year period, which appears in CESM1 as well (Fig. R7), was attributed to the westward propagation of baroclinic Rossby waves or the so-called thermal Rossby wave in the ocean (Frankcombe et al. 2008) wherein such propagation is produced by interactions between mean zonal advection, geostrophic self-advection, and ocean baroclinic Rossby waves (Sévellec & Fedorov 2013, 2015; Muir & Fedorov 2017; Ma et al. 2021). The

basin-wide baroclinic Rossby waves propagating in the ocean subsurface could be linked to an ocean-sea ice-atmosphere coupled mode (Ortega et al. 2015). Furthermore, role of ocean-atmosphere coupling in AMOC variability is also manifested in the feedbacks between the NAO and AMOC (Timmermann et al. 1998) or between the East Atlantic Pattern and AMOC (Msadek & Frankignoul 2009), or the combination of the two (Ruprich-Robert & Cassou 2015).

Thereupon, utilizing CESM1 to expound on the AMOC variability could lead to a new study including Rossby wave dynamics and ocean-atmosphere coupling, which is somehow beyond the scope of the current study. Given that the theories of AMOC variability have been well established, we reference the previous studies for mechanisms.

Zhang, J. & Zhang, R. On the evolution of Atlantic meridional overturning circulation fingerprint and implications for decadal predictability in the North Atlantic. *Geophys. Res. Lett.* **42**, 5419–5426 (2015).

Delworth, T., Manabe, S. & Stouffer, R. J. Interdecadal variations of the thermohaline circulation in a coupled ocean-atmosphere model. *J. Clim.* **6**, 1993–2011 (1993).

Griffies, S. M. & Tziperman, E. A Linear Thermohaline Oscillator Driven by Stochastic Atmospheric Forcing. *J. Clim.* **8**, 2440–2453 (1995).

Frankcombe, L., Dijkstra, H. & Von der Heydt, A. Sub-surface signatures of the Atlantic Multidecadal Oscillation. *Geophys. Res. Lett.* **35**, L19602 (2008).

Sévellec, F. & Fedorov, A. V. The leading, interdecadal eigenmode of the Atlantic meridional overturning circulation in a realistic ocean model. *J. Clim.* **26**, 2160–2183 (2013).

Sévellec, F. & Fedorov, A. V. Optimal excitation of AMOC decadal variability: links to the subpolar ocean. *Prog. Oceanogr.* **132**, 287–304 (2015).

Muir, L.C. & Fedorov, A. V. Evidence of the AMOC interdecadal mode related to westward propagation of temperature anomalies in CMIP5 models. *Clim. Dyn.* **48**, 1517–1535 (2017).

Ma, X., et al. Evolving AMOC multidecadal variability under different CO₂ forcings. *Clim. Dyn.* **57**, 593–610 (2021).

Ortega, P., et al. Reconciling two alternative mechanisms behind bi-decadal variability in the North Atlantic. *Prog. Oceanogr.* **137**, 237–249 (2015).

Timmermann, A., Latif, M., Voss, R. & Grötzner, A. Northern Hemispheric interdecadal variability: a coupled air–sea mode. *J. Clim.* **11**, 1906–1931 (1998).

Msadek, R. & Frankignoul, C. Atlantic multidecadal oceanic variability and its influence on the atmosphere in a climate model. *Clim. Dyn.* **33**, 45–62 (2009).

Ruprich-Robert, Y. & Cassou, C. Combined influences of seasonal East Atlantic Pattern and North Atlantic Oscillation to excite Atlantic multidecadal variability in a climate model. *Clim. Dyn.* **44**, 229–253 (2015).

iv) For completeness, in addition to the effect of the AMOC, sea ice is also an important forcing in the subpolar gyre warming/cooling. Can this effect be quantified in the model? This is one of the main uncertainties related to the AMOC spread in models, and since only one model was used here, it would be good to add this perspective. This analysis could be supplementary or in consonance with the previous point. Adding this analysis would considerably improve the knowledge of the effects of the ocean to the MHWs in the North Atlantic.

It is a very good suggestion. The sea ice effect has been included in the dynamic ocean. In the output of ocean component (POP) from the fully coupled CESM1 simulation, the melt

heat flux due to sea ice (MELTH_F) was included in the surface heat flux (SHF). Another variable QFLUX was also contained, which represents the internal ocean heat flux due to ice formation; heat of fusion > 0 or ice-melting potential < 0 . On the other hand, we exclusively employ CESM1 for the majority of the analyses because many other models did not provide daily SST outputs from both fully coupled and slab-ocean simulations. We will continue our study as new models make such outputs available in the future.

Minor comments:

L.33 "eastern tropical Pacific" for consistency throughout the manuscript.

Revised as suggested.

L. 99 "we see more frequent extreme ENSO episodes in dynamic ocean". Figure 1f seems to show otherwise, with the SST anomalies in the ENSO region being less frequent in the dynamic ocean. I agree with L. 102 that they are more intense and persistent.

We agree with the reviewer that Fig. 1f shows fewer MHW events in the dynamic ocean. More frequent extreme ENSO events tend to generate stronger and longer-lasting MHWs. When individual events become more persistent, they may occupy a larger fraction of the year. As a result, some events merge and appear as a single, persistent event in the event-counting algorithm and the annual-mean MHW frequency may appear reduced.

L. 131-133 easterly winds do not only produce westward advection, but also equatorial divergence and increase of vertical upwelling. This should be noted.

We have revised the text to clarify that the influence of easterly winds includes not only westward advection but also enhanced equatorial divergence and vertical upwelling, both of which contribute to local SST cooling.

L. 188 typo which "could" be understood.

Revised as suggested.

L. 193 Does this "positive anomalies" refer to a different phase of the variability or of a particular region in the map?

In this section, the "positive anomalies" refer to the warm SST anomalies that develop in the Gulf Stream region as part of the AMOC-related dipole pattern. When the AMOC weakens, the reduced northward heat transport generates negative SST anomalies south of Greenland, while the associated northward shift of the Gulf Stream produces positive SST anomalies (Caesar et al. 2018). These two centers form the characteristic AMOC fingerprint and similar to the spatial pattern of APT1.

Caesar, L., Rahmstorf, S., Robinson, A. Feulner, G. & Saba, V. Observed fingerprint of a weakening Atlantic ocean overturning circulation. *Nature* **556**, 191–196 (2018).

Reviewer #1 (Remarks to the Author):

The authors have responded to my review comments. Their key findings are that ocean dynamics can significantly enhance the intensity and prolong the duration of marine heatwaves in mid-to-high latitude oceans, and that marine heatwaves exhibit robust multi-year potential predictability. This is an important insight for improving heatwave prediction, given the frequent occurrence of such events. I recommend the acceptance of the manuscript.

Thank you very much for your positive feedback!

Reviewer #2 (Remarks to the Author):

This is my review of the revised version of the paper “Ocean dynamics shape marine heatwaves and their predictability”.

The authors addressed all my comments, including comparison with observations and acknowledging the caveats associated with this work. I have a few minor additional comments that I would like the authors to address before my final acceptance.

Thank you very much for your positive feedback!

Minor Comments:

L. 73: Has yet “to be” studied

Revised as suggested.

L. 88 ENSO power “spectrum”

Revised as suggested.

L. 126 We “compose the heat budget” or “make a composite of heat budget” not “we composite the heat budgets”

Revised as suggested.

L 228-231 - It is worth mentioning that ocean dynamics reduces the predictability of the ENSO region on interannual-to-decadal timescales.

Revised as suggested.

L. 338-352 The Bjerknes feedback terms contain several constants (alphas and betas). As I understand, these should be model dependent, representing for example the sensitivity of the ocean to wind forcing, and should be derived from these relationships. How were they derived? Please explain in the text.

We thank the reviewer for raising this point. The α and β coefficients in the Bjerknes stability index are model dependent. These coefficients are derived from linear regressions between SST anomalies and the associated oceanic and atmospheric response terms, thereby representing the sensitivity of ocean dynamics and air-sea coupling to SST variability. We have clarified this to the revised text.